# Post-Exercise Hypotension Induced by a Short Isometric Exercise Session Versus Combined Exercise in Hypertensive Patients with Ischemic Heart Disease: A Pilot Study

**DOI:** 10.3390/jfmk10020189

**Published:** 2025-05-25

**Authors:** Matteo Vitarelli, Francesco Laterza, Saúl Peñín-Grandes, Marco Alfonso Perrone, Alejandro Santos-Lozano, Maurizio Volterrani, Giuseppe Marazzi, Vincenzo Manzi, Elvira Padua, Barbara Sposato, Valentina Morsella, Ferdinando Iellamo, Giuseppe Caminiti

**Affiliations:** 1Department of Human Science and Promotion of Quality of Life, San Raffaele Open University, 00166 Rome, Italy; matteo.vitarelli@uniroma5.it (M.V.); maurizio.volterrani@uniroma5.it (M.V.); elvira.padua@uniroma5.it (E.P.); 2Department of Neurosciences, Biomedicine and Movement, University of Verona, 37134 Verona, Italy; 3Department of Wellbeing, Nutrition and Sport, Pegaso Open University, 80143 Naples, Italy; vincenzo.manzi@unipegaso.it; 4I+HeALTH Strategic Research Group, Department of Health Sciences, Miguel de Cervantes European University (UEMC), Padre Jiulio Chevalier Street, 2, 47012 Valladolid, Spain; spenin@uemc.es (S.P.-G.); asantos@uemc.es (A.S.-L.); 5Division of Cardiology and Sports Medicine, Department of Clinical Sciences and Translational Medicine, University of Rome Tor Vergata, 00133 Rome, Italy; iellamo@uniroma2.it; 6IRCCS San Raffaele, 00166 Rome, Italy; giuseppe.marazzi@sanraffaele.it (G.M.); barbara.sposato@sanraffaele.it (B.S.); valentina.morsella@sanraffaele.it (V.M.)

**Keywords:** isometric exercise, hypertension, ischemic heart disease, post-exercise hypotension

## Abstract

**Background:** Short sessions of isometric exercise have been shown to reduce blood pressure (BP) in normotensive and hypertensive subjects. However, there are few data in hypertensive patients with underlying ischemic heart disease (IHD). In the present study, we compared post-exercise hypotension (PEH) induced by isometric versus combined, aerobic plus dynamic resistance exercise in IHD patients. **Methods**: Twenty-five stable patients with established IHD and with treated hypertension were enrolled. The study had a cross-over design. All patients performed in a random order and on different days: (1) isometric exercise session (IES) consisting of bilateral knee extension, performed at 20% of maximal voluntary contraction and lasting 20 min; (2) combined exercise session (CES) including moderate-intensity continuous exercise at and dynamic resistance exercise performed at 60% of one repetition maximum, and lasting 60 min and (3) control session (no exercise). BP was measured at rest, immediately after the training and then every 15 min up to 90 min. **Results:** The repeated measures ANOVA analysis showed that systolic BP significantly decreased after the CES session compared to the control (F = 6.2; *p* 0.001) and IES (F = 4.4; *p* 0.004). Systolic BP significantly decreased after IES compared to the control (F = 3.6; *p* 0.036). Diastolic BP did not show significant changes after CES and IES compared to the control (CES vs. control: F = 2.2; *p* 0.142; IES vs. control (F = 2.5; *p* 0.062). There were no significant differences in diastolic BP changes between CES and IES (CES vs. IES: F = 1.8; *p* 0.156). **Conclusions:** We observed that CES was more effective than IES in reducing systolic BP; IES was as effective as CES in inducing diastolic PEH in hypertensive patients with underlying IHD.

## 1. Introduction

Exercise training is a well-established non-pharmacological intervention for the prevention and the treatment of hypertension [1,2]. In fact, almost all exercise modalities have been shown to reduce arterial blood pressure (BP) values with aerobic exercise having the most robust literature on this topic [3,4,5]. Isometric exercise (IE), consisting of muscle contractions in which the length of the muscle does not change, has also demonstrated arterial BP lowering effects both in normotensive and in hypertensive subjects [6,7]. Recent meta-analyses have reported that reductions in both systolic and diastolic BP, obtainable with IE training protocols, are comparable or superior to those of other forms of exercise and similar to that of standard anti-hypertensive medications [8]. In addition, the anti-hypertensive effects of IE are associated with a valuable time-efficiency and cost-sparing profile, and this makes IE a particularly attractive exercise modality in the long-term management of hypertensive patients [9]. However, IE is not currently recommended in the context of secondary prevention/cardiac rehabilitation programs due to the concern of causing excessive stimulation of the sympathetic nervous system leading to an abnormal rise in BP and, ultimately, inducing an abnormal increase in left ventricular filling pressure during the exercise [10,11]. Recent studies showed that there is a great variability regarding the hemodynamic response induced by IE, with low intensity small skeletal muscle mass involvement, such as during handgrips, and training status being mild or without an increase in systolic BP and LV filling pressure in patients with hypertension and underlying ischemic heart disease (IHD) [12,13,14]. It also emerges from the literature that performing handgrips at intensities that are too low, despite being hemodynamically safe, appears to be substantially ineffective in lowering BP in patients already treated with anti-hypertensive medications [15,16,17]. Therefore, the effort to define well-tolerated IE protocols should go hand in hand with checking whether they are effective or not in reducing BP in such patients. We recently showed that a single bout of isometric bilateral knee extension was well tolerated by patients with IHD and was more effective in reducing systolic BP than an isometric handgrip [14]. Post-exercise hypotension (PEH), defined as a transient reduction in BP following an exercise session, is a well-established method for testing the response of BP to a given type of exercise [18]. While IE has shown to induce significant PEH in hypertensive patients [19], there is a lack of data in those with underlying IHD. Considering the indications coming from the literature, we decided to assess the effects on systolic and diastolic BP of a single session of bilateral isometric knee extension in comparison to a session of combined, aerobic plus dynamic resistance exercise and to no exercise. The first aim of the study was to compare acute changes in systolic BP; the secondary endpoint was to compare acute changes in diastolic BP.

## 2. Materials and Methods

### 2.1. Population

The study included 25 hypertensive patients, consisting of 23 men and 2 women. The mean age of the entire population was 65.9 ± 8.2. All patients evaluated and included were attending a cardiac rehabilitation program. Inclusion criteria were as follows: age over 50 years and being in stable clinical conditions (no hospitalization in the last six month); having a stabilized pharmacological therapy (no changes in medications in the previous two months); having a diagnosis of IHD and being physically active. People were considered eligible if they had already carried out at least fifteen exercise sessions of their cardiac rehabilitation program or if they reported regular spontaneous home-based exercise (more than 2 sessions/week). Diagnostic criteria for IHD included the following: previous myocardial infarction (with or without ST-segment elevation) and previous coronary interventions (coronary artery bypass grafting and/or percutaneous coronary angioplasty). Exclusion criteria were as follows: incomplete coronary revascularization; myocardial ischemia or complex arrhythmias during the resting evaluation and/or during the ergometric test; previous diagnosis of chronic heart failure or presenting symptoms and/or signs of heart failure during the screening visit; documented ejection fraction below 40% and/or the ratio of E-wave velocity to the mean of left ventricular septal and lateral E-wave velocities (E/e’) above 14 at rest (at an echocardiography performed no more than three months before); permanent atrial fibrillation; resting baseline BP levels greater than 160/95 mmHg; anemia with hemoglobin levels below 10.5 g/dL; prior diagnosis of peripheral arterial disease with exercise-limiting claudication and advanced COPD (documented GOLD stage III or IV). Subjects with concomitant severe valvular heart disease and those with a diagnosis of hypertrophic cardiomyopathy were also excluded. All patients who met the inclusion criteria and agreed to participate in the study provided written informed consent before starting the study. This study complied with the Declaration of Helsinki and was approved by the local Ethics Committee of IRCCS San Raffaele Roma, Rome (protocol number 23/2024).

### 2.2. Study Design

The flow-chart of this research is reported in Figure 1. This was a pilot study with a cross-over design. It was conducted from September 2024 to February 2025 in the gym of San Raffaele IRCCS of Rome. Suitable patients were preliminarily evaluated by a cardiologist, a nurse and a physiotherapist in a screening visit, during which, the following examinations were carried out: collection of clinical history; measurement of resting heart rate (HR), resting systolic and diastolic BP and assessment of body mass index and waist circumference. A symptom-limited ergometric test was also performed during that visit. Patients fitting the inclusion/exclusion criteria were proposed to participate in the study. Patients who agreed, in order to familiarize with the experimental protocol, were summoned to a trial session during which they tried out the devices and exercises that were the object of the study. Afterward, every patient completed three experimental sessions: the (1) isometric exercise session (IES); (2) combined session (CES) and (3) control session. Experimental sessions were carried out on different days separated by >72 h. The order of these sessions was randomly assigned through a computer software.

### 2.3. Exercise Settings

Ergometric test: the test performed during the screening visit was used to set the intensity of the aerobic component of the combined exercise session. Patients performed an incremental stepwise test on a cycle ergometer (Quark CPET, Cosmed, Frascati, Italy). The exercise was initiated at a load of 20 W, and the load was increased every two minutes by 20 W. The pedaling frequency was set at around 60 revolutions per minute during the entire exercise; the exercise was interrupted when volitional exhaustion occurred. There was continuous monitoring of HR during the exercise test, while BP was measured at rest, at every stage of the incremental exercise and at the first and the third minute of recovery. At each stage the rate of perceived exertion (RPE) was acquired by using the Borg’s scale [20].

The test for determining maximal voluntary contraction (MVC) consisted of 3 maximal contractions, each one lasting 3–5 s each, with 1 min of rest between contractions [21]. An evaluator encouraged patients, during each contraction, to exert their maximal effort. For each trial, the maximum force generated by the patient was recorded; the maximum value over the three trials was used as the final measurement. The test for assessing the 1-repetition maximum (1RM) was performed according to a standard protocol [22]. During the test, patients were first instructed to complete a warm-up set comprising eight repetitions at 50% and six repetitions at 70% of their perceived 1RM. The load was then progressively increased until reaching the workload that could be lifted between three and five times (3–5 RM), with 2–3 min rest between efforts. The 1RM workload was determined for the chest press, vertical pull, leg extension and leg press. The MVC and 1RM tests were repeated on another 2 days to check for reproducibility. The intraclass correlation coefficient was used to measure reproducibility, with a good agreement designated as >0.80 [23].

### 2.4. Experimental Sessions

All experimental sessions were carried out in the cardiac rehabilitation gym of San Raffaele IRCCS of Rome. They were performed in the morning, between 8:30 a.m. and 10.30 a.m. Patients were instructed to refrain from alcohol consumption and cigarette smoking at least 48 h before the experimental sessions. They were also asked to avoid physical activity for 24 h before each session. A light breakfast was allowed at least 2 h before the sessions. Exercise sessions were preceded by 15 min of pre-session rest and followed by 90 min of post-session rest. During the pre-session rest, patients remained seated in an armchair in a quiet environment. Systolic and diastolic BP and HR were measured every five minutes, and their average values were considered as baseline measurements.

Isometric exercise sessions (IES) lasted 20 min and were performed on a knee flex/extension dynamometer (Technogym Wellness System, Technogym, Cesena, Italy). Participants were seated on the dynamometer; the seat was adjusted for each individual so that the axis of rotation around the dynamometer shaft was adjacent to the lateral femoral condoyle of the patient’s right leg. Patients had their knees bent at a 90° angle. Both legs were positioned underneath the knee extension/flexion attachment arm of the dynamometer. The intensity of the exercise was set at 20% of the MVC. Every session lasted 20 min and was structured as follows: patients performed 5 sets of exercise, with each set lasting 2 min. There were two-minute rest periods between sets.

Combined exercise sessions (CES) lasted for 60 min overall and consisted of 40 min of aerobic exercise and 20 min of dynamic resistance exercise. The aerobic component of the session consisted of continuous exercise that was alternatively carried out on a treadmill or a bike, according to the patients’ preference. The intensity of the aerobic component was established by means of the RPE method with an intensity target of 13–14 (somewhat hard). The dynamic resistance component of the CES included the chest press, vertical pull, leg extension and leg press. For each of these exercises, patients performed 3 sets of 12 repetitions with a one-minute of rest between the sets. The intensity of the exercise was set at 60% of 1RM.

Post-session rest: patients remained seated in an armchair in a quiet environment. BP values were recorded immediately post exercise session (PE), and then every 15 min up to 90 min (15, 30, 45, 60, 75 and 90). An electronic device (Omron 705, Milan, Italy) was used to record BP values. All exercises of the CES were performed with Technogym machines (Wellness System, Technogym, Cesena, Italy). During control sessions (CS), patients were asked to stay at rest, sitting on an armchair, for 120 min.

### 2.5. Statistical Analysis

This research was conceived as a pilot study; no formal a priori power analysis was performed. Therefore, the sample size was determined by feasibility and patients’ availability. The data were submitted to the Shapiro–Wilk test for normality. Two repeated measure ANOVAs were conducted to assess changes in systolic and diastolic blood pressure across time and experimental sessions. Time (baseline and post-exercise) and session (treatment vs. control condition) were included as within-subject factors. Bonferroni’s post hoc tests were performed if needed. The absolute values for SBP and DBP were converted to a percent variation in rest (Δ%) and treated the same way. The significance level was set at *p* < 0.05. SPSS v26 was used for conducting all statistical analyses (*p* < 0.05). The data were analyzed using SPSS software (version 26.0 IBM Corp., Armonk, NY, USA).

## 3. Results

Out of the initial number of 73 patients recruited, 48 were excluded. In total, 32 out of 48 (66%) patients did not meet the inclusion/exclusion criteria while 16 declined to participate in the study. The reason for refusing the participation was the lack of willingness to perform the experimental sessions. Twenty-five patients were then randomized. All patients included and randomized completed the study protocol and their data were analyzed. Characteristics of the study population are reported in Table 1. The EF ranged from 40 to 58%. In total, 9 (36%) out of 25 patients had EF below 50%. A total of 18 (75%) out of 25 patients had a previous myocardial infarction. Patients were taking an average of 2.6 ± 1.4 drugs for BP control. In total, 6 (24%) of the 25 patients were obese (with BMI over 30 kg/m^2^) and 14 (56%) were overweight (with BMI between 25 and 30 kg/m^2^). All patients tolerated the exercise sessions well and none developed symptoms.

When compared to baseline values, the average PEH for systolic BP was −8.4 ± 2.1 mmHg and −3.7 ± 1.6 mmHg after CES and IES session, respectively. The greater reduction in systolic BP was observed at 30 min of recovery after CES (−10.5 ± 3.6 mmHg) and at 45 min of recovery after IES (−5.7 ± 2.4 mmHg). The average PEH for diastolic BP was −2.1 ± 1.3 mmHg and −2.4 ± 0.8 mmHg, after CES and IES sessions, respectively. The greater reduction in systolic BP was observed at PE (−4.4 ± 1.9 mmHg) after CES and at 45 min of recovery (−2.7 ± 1.3 mmHg) after IES. The repeated measures ANOVA revealed an interaction between measurement time and sessions for systolic BP but not for diastolic BP (Figure 2 and Figure 3). Systolic BP significantly decreased after CES compared to the control (F = 6.2; *p* 0.001) and to IES (F = 4.4; *p* 0.004). Systolic BP significantly decreased after IES compared to the control (F = 3.6; *p* 0.036). Post hoc analyses showed that systolic BP decreased significantly after CES compared to IES and the control at PE, and 90 min, and compared to control at 15, 30, 45, 60 and 75 min. After IES, systolic BP significantly decreased at 15′, 30′ and 45′ compared to the control. Diastolic BP did not show a significant decrease after CES and IES compared to the control (CES vs. control: F = 2.2; *p* 0.142; IES vs. control (F = 2.5; *p* 0.062). There were no significant differences in diastolic BP changes between CES and IES (CES vs. IES: F = 1.8; *p* 0.156).

## 4. Discussion

There is a growing body of literature demonstrating the effectiveness of IE in reducing BP in both normotensive and hypertensive individuals. However, the potential use of IE as an adjuvant therapy for managing hypertension in patients with ischemic heart disease IHD remains largely unexplored. The present study yielded two main findings. First, a short IES, consisting of bilateral knee extension, induced a significant systolic PEH in hypertensive patients with underlying IHD, although it was less effective than a longer conventional CES. Second, the IES and CES produced a similar not-significant reduction in diastolic BP. It is noteworthy that the IES was shorter in duration compared to the CES (20 min vs. 60 min). This asymmetrical design was deliberately chosen by the authors to investigate the effectiveness of a more time-efficient exercise modality, with the aim of facilitating its future integration into the daily routine of hypertensive patients with IHD. The duration of the IE session in this study was consistent with previous research, where IE sessions typically ranged from approximately 11 to 20 min [8,24]. In contrast, conventional exercise modalities commonly used in cardiac rehabilitation—such as aerobic training, either alone or in combination with dynamic resistance exercises—usually last from 30 min to over one hour. In this study, the intensity of isometric bilateral knee extension was set at 20% of MVC, aligning with most prior studies that employed low-intensity IE protocols [8]. Although the magnitude of the PEH response appears to increase linearly with IE intensity [25], the choice of a low-intensity protocol was justified to avoid excessive sympathetic nervous system activation during exercise. Exercise intensity is a critical variable in prescribing IE training. It requires balancing the hypotensive effects with the need to avoid excessive increases in systolic BP during exercise, which could be particularly harmful in patients with existing cardiac damage, such as those with IHD. High-intensity IE may provoke substantial rises in systolic BP and a corresponding increase in left ventricular (LV) filling pressure—an undesirable effect in this patient population [26]. In a previous study, our group demonstrated that bilateral isometric knee extension at 20% MVC was well tolerated by physically active hypertensive patients with IHD; despite a significant rise in systolic BP during the isometric effort, echocardiographic measures of LV filling pressure remained unchanged [14]. Regarding the BP-lowering effects, the findings of the present study differ slightly from those reported in the literature. For instance, a recent meta-analysis by Inder et al. [27] showed that IE training significantly reduced systolic and diastolic BP, with mean differences of −5.20 mmHg and −3.90 mmHg, respectively. However, direct comparisons should be approached with caution for at least two key reasons. First, most available data on the acute and long-term effects of IE on BP have been derived from normotensive or hypertensive patients without cardiac disease [28]. In contrast, few studies have specifically examined complex patients with IHD who are often on multiple BP-lowering medications. Second, the studies included in various meta-analyses are heterogeneous in terms of both exercise intensity and the specific IE protocols used. For example, most IE research has utilized handgrip protocols, typically performed at 30% of the participant’s MVC. The results of the present study are consistent with other research utilizing low-intensity isometric knee extension protocols [29,30,31], and partially agree with the study of Ash et al. [32] showing that aerobic exercise was more effective than isometric exercise in reducing systolic BP. Conversely, they differ from those of Oliveira et al. [19] who found no reduction in BP within 60 min post-exercise, either after isometric bilateral knee extension or handgrip exercise in hypertensive patients. The difference between the two studies may be due to the different protocols used and the underlying pharmacological therapy. Oliveira et al. employed an intensity of 30% of maximal strength and one-minute intervals between contractions. This may have elicited a greater sympathetic response compared to the present study, where the contraction intensity was lower and the rest intervals between contractions were two minutes. Moreover, the systematic use of betablockers among patients of the present study may have attenuated the sympathetic nervous system activation during the post-exercise phase. Ultimately, this research supports the use of short IE sessions for the management of hypertension in IHD patients. We think that the results of this study, if confirmed and expanded by future studies, could contribute to clarify the role of IE in the context of the non-pharmacological management of hypertension, and could offer new perspectives for the treatment of hypertension in IHD patients.

For limitations, we measured BP for 90 min from the end of the exercise phases. Therefore, this study did not provide information on the duration of the PEH effect; further studies, preferably using a 24/BP monitoring approach, are needed to confirm and expand our results. The study employed an asymmetrical approach, as the duration of the IES was significantly shorter than that of the CES. This may have led us to underestimate the blood pressure-lowering effect of interval exercise. Ideally, for a proper comparison between two different exercise modalities, the duration of the sessions should be matched. However, the primary aim of the present study was to assess the effectiveness of a short session of IE in our population. The authors’ ultimate goal was to identify a minimal “dose” of exercise that could be easily repeated on a daily basis by patients with IHD since we believe that the smaller the required exercise dose, the greater the likelihood of long-term adherence. In this context, the present study, representing the first step in a broader project, has enabled authors to better understand the magnitude of PEH induced by IE, in comparison to both no exercise and the gold standard intervention. The study enrolled only two women, and this low representation of females limits the generalizability of the study results to the female population. Despite the very low representation of the female gender, the authors decided to include the female participants in the analysis in order to preserve the original design of the study. The study assessed the acute BP response to IE in IHD patients. Although it has been shown that the acute response to IE predicts the long-term antihypertensive effectiveness of isometric training [33], further studies are needed to assess the medium- and long-term antihypertensive effects of this exercise modality. In the present study IE was performed as bilateral knee extension; this kind of exercise requires the availability of a dynamometer and a certain degree of skill on the part of the patient; therefore, it is not well suited for long-term treatments outside the rehabilitation institute setting. Further studies should investigate whether the PEH effects observed with bilateral knee extension in the present study are comparable to those of other types of isometric exercise that are more suitable for long-term, home-based treatment.

## 5. Conclusions

The results of the present study suggest that IE, performed as bilateral knee extension, may be effective in eliciting systolic PEH in patients with hypertension and underlying IHD. Further studies are needed in order to prove whether IE is a feasible non-pharmacological intervention for the long-term management of hypertension in these patients.

## Figures and Tables

**Figure 1 jfmk-10-00189-f001:**
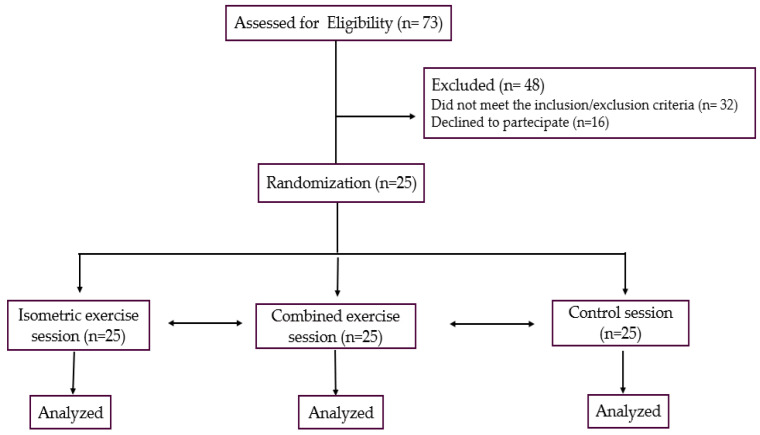
Study flow-chart.

**Figure 2 jfmk-10-00189-f002:**
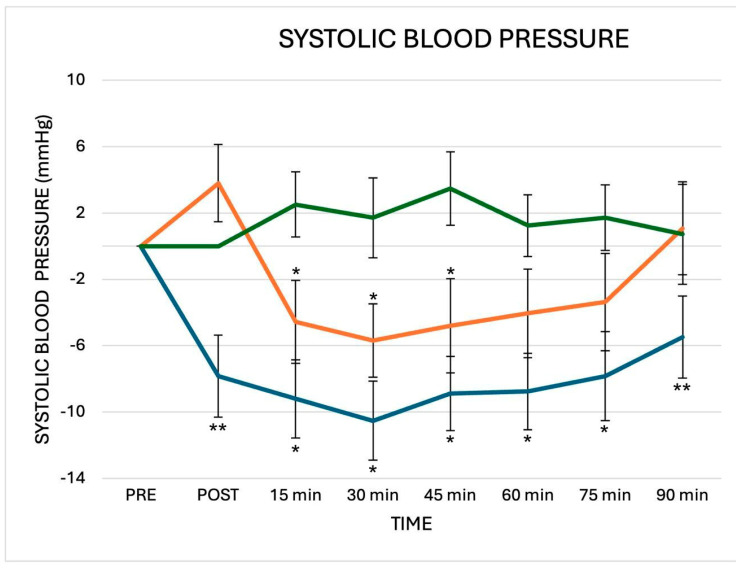
Changes in systolic BP observed after the three experimental sessions: CES (blue line), IES (orange line) and control (green line). Delta = post-exercise points vs. pre-exercise. ** *p* < 0.05 vs. active session and control; * *p* <0.05 vs. control. The statistical analysis was performed by repeated measures ANOVA and Bonferroni’s post hoc tests.

**Figure 3 jfmk-10-00189-f003:**
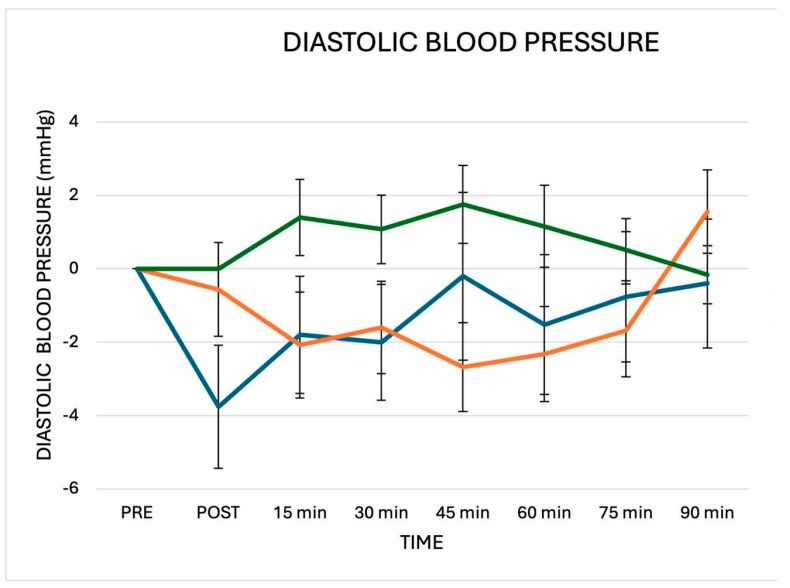
Changes in diastolic BP observed after the three experimental sessions: CES (blue line), IES (orange line) and control (green line). Delta = post-exercise points vs. pre-exercise. No significant between-sessions differences were detected. The statistical analysis was performed by repeated measures ANOVA.

**Table 1 jfmk-10-00189-t001:** Anthropometric and clinical features of patients included in the study.

Age, years	65.9 ± 8.2.
BMI, kg/m^2^	28.1 ± 8.2
Waist circumference, cm	106.2 ± 31.6
Male/female, n	23/2
Previous PCI/CABG, n	19/10
EF, (%)	52.7 ± 6.7
NT-proBNP, ng/pL	124.2 ± 31.6
Comorbidities	
Carotid artery disease, n (%)	14 (56)
Diabetes, n (%)	7 (24)
Hypercholesterolemia, n (%)	22 (88)
Previous Smoke habit, n (%)	15 (60)
Ergometric test	
Time of exercise, s	295.8 ± 87.3
MET	5.8 ± 1.6
Treatment	
Anti-platelets agents, n (%)	25 (100)
ACE-Is/ARBs, n (%)	22 (88)
Betablockers, n (%)	21 (84)
CCBs, n (%)	11 (44)
SGLT2-I, n (%)	7 (28)
Diuretics, n (%)	8 (32)
Statins, n (%)	25 (100)

BMI = body mass index; PCI = percutaneous coronary intervention; CABG = coronary artery bypass grafting; EF = ejection fraction. ACE-Is = angiotensin converting enzyme inhibitors; ARBs = angiotensin receptor blockers; CCBs = Calcium channel blockers.

## Data Availability

The data presented in this study are available upon request from the corresponding authors.

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
