# Peer review of "Post-Exercise Hypotension Induced by a Short Isometric Exercise Session Versus Combined Exercise in Hypertensive Patients with Ischemic Heart Disease: A Pilot Study"

_jfmk, 2025, doi:10.3390/jfmk10020189_

Round 1
Reviewer 1 Report
Comments and Suggestions for Authors
I think the major issue in this study is that isometric exercise only comprised legs and was 20 min long. In contrast, the combined group activated much more muscle regions and the training duration was 3 times higher (60 min). Hence, direct comparison is difficult.
Further, the studies that showed that isometric exercise training such as wall-sit induced blood pressure lowering were performed over 12 weeks. Hence, a single reduction in blood pressure after one session is quite notable.
Overall, the study bespeaks the additonal value of isometric exercise in patient with atherosclerotic coronary disease.
Author Response
I think the major issue in this study is that isometric exercise only comprised legs and was 20 min long. In contrast, the combined group activated much more muscle regions and the training duration was 3 times higher (60 min). Hence, direct comparison is difficult.
Thank you for this comment
We agree with the reviewer that, for a proper comparison between two different exercise modalities, it would be ideal to match the duration of the sessions. However, the primary aim of the present study was to evaluate the effectiveness of a short session of interval exercise in our population. Our ultimate goal is to identify a minimal “dose” of exercise that can be easily repeated on a daily basis by these patients. We believe that the smaller the required exercise dose, the greater the likelihood of long-term adherence. In this sense, the present study, that represents a first step of that project, has allowed us to understand the magnitude of post-exercise hypotension induced by interval exercise, in comparison to both no exercise and the gold standard intervention. In a way, it addresses the following question: is it worthwhile for patients with IHD to perform IE in order to reduce blood pressure? We have now added this comment among study limitations.
Further, the studies that showed that isometric exercise training such as wall-sit induced blood pressure lowering were performed over 12 weeks. Hence, a single reduction in blood pressure after one session is quite notable.
Thank you for this comment
We agree with the reviewer’s comment regarding the need of middle or long-term studies in order to better understand the potential role of IE in hypertensive patients with IHD . This aspect has already been addressed in the study’s limitations:
“The study assessed the acute BP response to IE in IHD patients; although it has been shown that the acute response to IE predicts the long-term antihypertensive effectiveness of isometric training [33], further studies are needed to assess the medium- and long-term antihypertensive effects of this exercise modality”
Reviewer 2 Report
Comments and Suggestions for Authors
Dear Authors,
After reviewing your manuscript, I present below a series of comments and suggestions in order to optimize its presentation and clarity.
Introduction: It would be advisable for the objective of the study to be explicitly stated at the end of the introduction.
Methodology:
- Equipment: The manuscript mentions "Quark CPET" after "cycle ergometer." It is important to clarify this sequence, as the Quark CPET is the gas analyzer used to measure variables during the cycle ergometer test, not a subsequent component or type of ergometer.
- Variables: The recording and reporting of maximal oxygen consumption (VO2max) values are absent. Could you please clarify why these data, a fundamental variable in cardiorespiratory fitness studies, were not included?
- Statistical Analysis: It is necessary to cite the bibliographic source that supports the assertion that an intraclass correlation coefficient (ICC) of 0.80 is considered a "good value" or an appropriate threshold for reliability.
- Please ensure that the letter "p" referring to the p-value is consistently presented in italics (e.g., p < 0.05).
- The manufacturer's company name and location (city, country) for the SPSS software used should be specified (e.g., SPSS Inc., Chicago, IL, USA).
Results
- Tables: Please review and correct the format of the table(s) to ensure clarity and compliance with the journal's guidelines.
- Figures: The font type and size used in figures/images should be consistent with that of the main manuscript text and/or adhere to the journal's style guidelines. Please correct the description in the caption for Figure 2, specifically the phrase or information following the mention of "blue..." (or the corresponding color if "blue" was an example).
- Data Presentation: A space should be included before and after the ± symbol when presenting results (e.g., "value ± SD").
- In the Results section, two consecutive sentences begin with the same phrasing: "When compared to baseline values...". It is recommended to vary the wording to improve text flow.
Discussion: Given the small number of female participants (n=2), the question arises whether it would have been methodologically more robust or interpretively clearer to analyze and present only the results for the male participants, despite the consequent reduction in sample size. Alternatively, the inclusion of these two participants in the combined analysis should be further justified, and the potential impact of this heterogeneity on the results should be discussed.
Author Response
Introduction: It would be advisable for the objective of the study to be explicitly stated at the end of the introduction.
Thank you for this comment. In the revised version of the manuscript we stated the objective of the study
Methodology:
- Equipment: The manuscript mentions "Quark CPET" after "cycle ergometer." It is important to clarify this sequence, as the Quark CPET is the gas analyzer used to measure variables during the cycle ergometer test, not a subsequent component or type of ergometer.
Thank you for this comment. Our version of the Quark CPET allows to perform ergometric test without performing gas analysis.
Variables: The recording and reporting of maximal oxygen consumption (VO2max) values are absent. Could you please clarify why these data, a fundamental variable in cardiorespiratory fitness studies, were not included?
Data regarding VO2max were not available since our patients performed an ergometric test without gas analysis. According to the reviewer suggestion, in table 1 we added METs reached during the ergometric test and time of exercise
Statistical Analysis: It is necessary to cite the bibliographic source that supports the assertion that an intraclass correlation coefficient (ICC) of 0.80 is considered a "good value" or an appropriate threshold for reliability.
Thank you. In the revised version of the manuscript we have added a bibliographic source that supports the assertion that an intraclass correlation coefficient (ICC) of 0.80 is considered a "good value"
Please ensure that the letter "p" referring to the p-value is consistently presented in italics (e.g., p < 0.05).
Thank you. Done
The manufacturer's company name and location (city, country) for the SPSS software used should be specified (e.g., SPSS Inc., Chicago, IL, USA).
Results
Tables: Please review and correct the format of the table(s) to ensure clarity and compliance with the journal's guidelines.
Thank you. Done
Figures: The font type and size used in figures/images should be consistent with that of the main manuscript text and/or adhere to the journal's style guidelines. Please correct the description in the caption for Figure 2, specifically the phrase or information following the mention of "blue..." (or the corresponding color if "blue" was an example).
Thank you. We have implemented figures according to the journal's style guidelines. We corrected that phrase in caption of figure 2
Data Presentation: A space should be included before and after the ± symbol when presenting results (e.g., "value ± SD").
Thank you. Done
In the Results section, two consecutive sentences begin with the same phrasing: "When compared to baseline values...". It is recommended to vary the wording to improve text flow.
Thank you, we deleted the repeated sentence “When compared to baseline values” from page 6, line 217.
Discussion: Given the small number of female participants (n=2), the question arises whether it would have been methodologically more robust or interpretively clearer to analyze and present only the results for the male participants, despite the consequent reduction in sample size. Alternatively, the inclusion of these two participants in the combined analysis should be further justified, and the potential impact of this heterogeneity on the results should be discussed.
Thank you for this comment. The authors took these aspects into consideration during the data analysis phase. However, they decided to include the two female participants in the analysis, as the results remained unchanged when comparing analyses with and without them. This also allowed to preserve the original design of the study.